# Phage display and selection of lanthipeptides on the carboxy-terminus of the gene-3 minor coat protein

Johannes H. Urban[1], Markus A. Moosmeier[1], Tobias Aumüller[1], Marcus Thein[1], Tjibbe Bosma[2], Rick Rink[2], Katharina Groth[1], Moritz Zulley[1], Katja Siegers[1], Kathrin Tissot[1], Gert N. Moll[2] & Josef Prassler[1]

Ribosomally synthesized and post-translationally modified peptides (RiPPs) are an emerging class of natural products with drug-like properties. To fully exploit the potential of RiPPs as peptide drug candidates, tools for their systematic engineering are required. Here we report the engineering of lanthipeptides, a subclass of RiPPs characterized by multiple thioether cycles that are enzymatically introduced in a regio- and stereospecific manner, by phage display. This was achieved by heterologous co-expression of linear lanthipeptide precursors fused to the widely neglected C-terminus of the bacteriophage M13 minor coat protein pIII, rather than the conventionally used N-terminus, along with the modifying enzymes from distantly related bacteria. We observe that C-terminal precursor peptide fusions to pIII are enzymatically modified in the cytoplasm of the producing cell and subsequently displayed as mature cyclic peptides on the phage surface. Biopanning of large C-terminal display libraries readily identifies artificial lanthipeptide ligands specific to urokinase plasminogen activator (uPA) and streptavidin.

[1] MorphoSys AG, Semmelweisstr. 7, 82152 Planegg, Germany. [2] Lanthio Pharma, Rozenburglaan 13 B, 9727 DL Groningen, Netherlands. Johannes H. Urban and Markus A. Moosmeier contributed equally to this work. Correspondence and requests for materials should be addressed to J.H.U. (email: johannes.urban@morphosys.com)

The high degree of chemical and structural diversity found in ribosomally synthesized and post-translationally modified peptides (RiPPs) and other natural products currently sparks the interest in mining and engineering these peptides for drug discovery[1]. Among them, lanthipeptides, a subclass of RiPPs with more than 100 members described to date, are produced in a wide range of bacteria[2] and are characterized by the presence of enzymatically introduced thioether-bridged amino acids, called lanthionines (Lan) and methyllanthionines (MeLan). In a first enzymatic reaction during biosynthesis, a dehydratase recognizes an N-terminal leader sequence within the LanA precursor peptide and catalyzes the dehydration of serines and threonines in the core peptide to 2,3-didehydroalanine (Dha) and 2,3-didehydrobutyrine (Dhb), respectively. Subsequently, a cyclase supports the addition of cysteine thiols to the unsaturated amino acids Dha and Dhb to form the covalent Lan and MeLan linkages. Whereas dehydration and cyclization of class I lanthipeptides are catalyzed by separate LanB and LanC enzymes, respectively, a single bifunctional dehydratase/cyclase termed LanM is responsible for the orchestrated thioether formation in class II lanthipeptides. The thioether linkages confer thermal and proteolytic stability, frequently increase the affinity to protein targets by reducing the conformational flexibility of the peptide structure, and in contrast to disulfides, are resistant to chemical reduction[3]. Most lanthipeptides characterized so far have antimicrobial activity and are collectively termed lantibiotics. However, screening of microbial strain collections or purified peptides[4–6] recently identified lanthipeptides with unanticipated antifungal, antiviral, and anti-allodynic bioactivities and genome-mining efforts uncovered novel variants that await characterization[7, 8]. The recent advance in lanthipeptide research culminated in the exploitation of heterologous hosts, e.g., *Escherichia coli* (*E. coli*) for improved expression[9, 10], the in vitro reconstitution of several biosynthetic enzymes[11, 12], and the successful solid-phase synthesis of even complex lanthipeptides with intertwined ring structures[13–15]. Despite this progress, the systematic engineering of lanthipeptides is still in its infancy[16]. Nonetheless, a proof-of-concept lanthipeptide cell display system based on *Lactococcus lactis* (*L. lactis*) was recently described[17]. Unfortunately, the poor transformation efficiency of Gram-positive bacteria, the lack of a robust genetic engineering toolbox, and a tendency toward cell-clumping strictly limits this approach. Another study engineered artificial lanthipeptides by messenger RNA display in a Lan enzyme free system, which is based on chemical thioether bridge introduction via a reactive non-proteinogenic amino acid[18]. In contrast to enzymatic thioether formation, the major drawback of this rather cumbersome method is the lack of stereo- and regioselectivity of cycle formation that requires downstream deconvolution.

We envisioned to use *E. coli* as heterologous host for the production of large enzymatically modified lanthipeptide libraries to be displayed on the surface of the filamentous bacteriophage M13 and to enable the screening of peptide ligands to targets of choice. Phage display is a robust and widely used in vitro selection technology for the discovery of therapeutic peptides and antibodies[19]. For the successful display of lanthipeptides on phage several requirements must be met: (i) heterologous co-expression of the biosynthetic enzymes along with a genetic fusion comprising the cognate leader peptide, a core peptide library, and a phage coat sequence, (ii) efficient recognition of the leader peptide by the modifying enzymes and modification of the peptide core, both in the context of the fusion protein, (iii) incorporation of the fusion into the inner membrane of *E. coli* prior to phage assembly, and (iv) assembly and release of infectious particles displaying the cyclic, thioether-bridged peptides.

Here, we identified the rarely used carboxy-terminus of the gene-3 minor coat protein (pIII) to be ideally suited for the display and de novo selection of lanthipeptides. C-terminal precursor peptide fusions to pIII ensure prolonged interaction with the heterologous Lan enzymes expressed in the cytoplasm of the producer cell prior to phage assembly and support the display of extensively post-translationally modified peptides on the phage surface. Using this C-terminal display mode on pIII, we were able to select cyclic lanthipeptide ligands to uPA and streptavidin from phage display libraries. We believe that phage display on the C-terminus of pIII is an excellent tool for the mining of novel lanthipeptides and might be suitable for the engineering of other RiPP classes that rely on leader peptide-dependent enzymatic modification.

## Results

**An ELISA-based reporter assay monitors peptide cyclization.** To adapt phage display for the engineering of lanthipeptides, we focused on the class I biosynthetic NisBC system from *L. lactis* and the class II ProcM system from *Prochlorococcus* MIT9313, both known to support heterologous expression of lanthipeptides in *E. coli*[10]. While the well-characterized NisBC two-enzyme system acts on the NisA precursor peptide as sole natural substrate to synthesize the antimicrobial lantibiotic nisin[20], the bifunctional ProcM turns 29 ProcA precursors with conserved

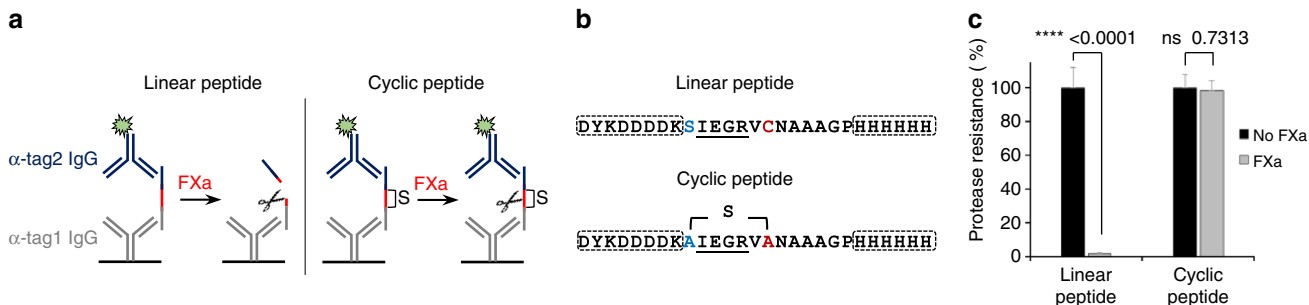

**Fig. 1** An ELISA-based reporter assay monitors the cyclization status of artificial lanthipeptides. **a** Peptides containing an FXa recognition site (red bar) flanked by affinity tags and residues involved in thioether bridge formation are captured via anti-tag1 antibodies, treated with FXa, and detected via anti-tag2 antibodies. Tag2 is proteolytically removed (left panel) in linear peptides, but remains connected via a covalent thioether (indicated by "S", right panel) in cyclic peptides, resulting in low and high signals, respectively. **b** Sequences of synthetic linear or thioether-bridged peptides with FLAG- and His$_6$-epitopes (boxed) flanking the FXa site (underlined) used for assay validation. **c** Synthetic peptides were captured via the His$_6$-tag, incubated with or without FXa, and detected using anti-FLAG IgG. The protease resistance relative to untreated (no FXa) samples was calculated and data representing mean ± s.d. of three replicates is shown (unpaired, two-tailed *t*-test). The experiment was repeated three times

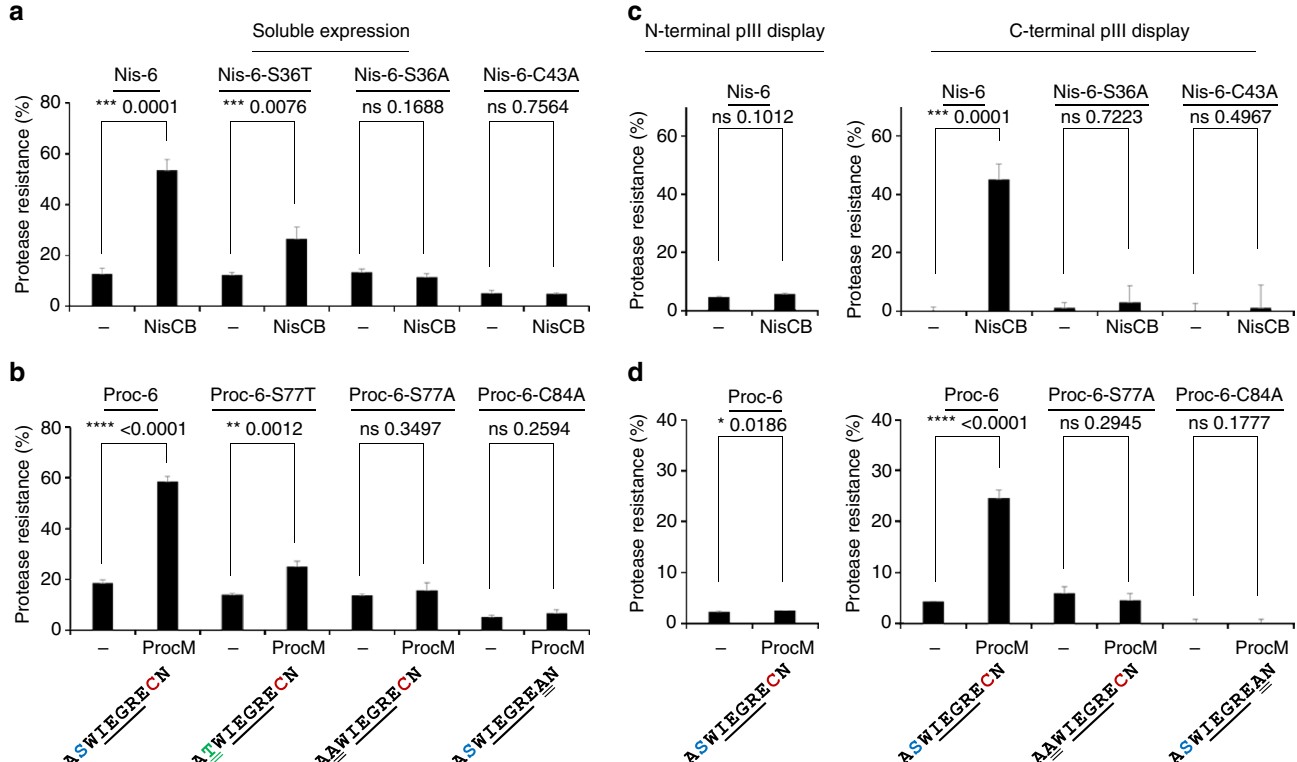

**Fig. 2** Assessment of the cyclization status of artificial lanthipeptides in cell lysates and displayed on phage. **a** Precursor peptides (full sequences in Supplementary Table 1) containing the NisA leader and indicated core sequences (residues involved in thioether formation colored) flanked by affinity tags were expressed with or without NisBC, captured from cell lysates, and subjected to FXa digestion and ELISA detection. The protease resistance relative to untreated (no FXa) samples was calculated and data representing mean ± s.d. of three independent cultures analyzed in duplicate is shown (unpaired, two-tailed *t*-test). **b** As in **a**, but core sequences were fused to the ProcA leader (full sequences in Supplementary Table 2) and expressed with or without ProcM enzyme. **c** Peptides containing the NisA leader sequence were translationally fused to the N- (left panel) or C-terminus (right panel) of phage pIII coat protein (full sequences in Supplementary Table 3), phage produced with or without NisBC co-expression, and the peptide cyclization status assessed on phage particles as described in **a**. **d** As in **c**, but phage displayed peptides containing ProcA leader sequences (full sequences in Supplementary Table 4) were tested after phage production with or without ProcM co-expression. Experiments shown in **a**–**d** were repeated three times

leader, but unrelated core peptide sequences into structurally diverse prochlorosins of unknown function[21, 22]. Since the unambiguous analytical verification of thioether bridge formation is cumbersome, especially in the context of phage particles, we developed an ELISA-based reporter assay that allows to rapidly monitor the cyclization status of model peptides in cell lysates or displayed on phage (Fig. 1a). Herein, a factor Xa (FXa) protease cleavage site is flanked by a serine or threonine and a cysteine, the residues involved in enzymatic thioether formation, and two affinity tag sequences which enable ELISA-based capture and detection. Synthetic linear or thioether-bridged peptides of related sequence (Fig. 1b) were tested in a sandwich-ELISA and confirmed that FXa treatment of linear peptides leads to proteolytic separation of the affinity tags and signal loss, whereas signals after FXa treatment of cyclic peptides are unaffected since the tags remain connected via the covalent thioether linkage (Fig. 1c).

**Expression and phage display of artificial lanthipeptides.** With this reporter system at hand, a series of FXa site containing peptide variants were fused to the natural NisA and ProcA leader peptides, heterologously expressed in *E. coli* along with the modifying Lan enzymes, and subsequently cell lysates were screened for peptide cyclization. We identified NisA reporter peptides containing an FXa site flanked by serine/cysteine residues and FLAG and His6-epitopes that are largely resistant to FXa treatment when co-expressed with the modifying NisBC enzymes,

but are readily proteolytically cleaved in absence of the Lan enzymes (Fig. 2a). Since enzymatic dehydration of serine to the reactive Dha can result in spontaneous cyclization, a serine to threonine point mutation was introduced, which is more chemically inert after dehydration to Dhb[23]. Even though the proteolytic resistance was reduced, the threonine-containing mutants still revealed a significant fraction of FXa-resistant peptides when NisBC was co-expressed. Mutating the serine or cysteine residues required for thioether bridge formation to alanine, abrogated FXa cleavage resistance even with NisBC co-expression. The same reporter core peptide sequence fused to the ProcA leader and co-expressed with ProcM enzyme did not result in FXa-resistant peptides. However, when the highly charged FLAG-tag was replaced by a more charge neutral HA-tag enzymatic cyclization of the core peptide sequence, as judged by FXa cleavage resistance, was observed (Fig. 2b). Again, mutation of the serine to threonine reduced the cyclization efficacy, whereas cyclization was abrogated when either serine or cysteine was mutated to alanine. Next we set out to display the identified reporter peptides on phage particles and to assess their cyclization status. A phagemid was constructed that encodes the precursor peptide sequences equipped with an *ompA* signal sequence for periplasmic transport fused to the N-terminus of the pIII phage coat protein. Phage particles produced with or without co-expression of the cognate Lan enzymes were captured onto ELISA plates using anti-pVIII antibodies and subjected to FXa cleavage assays. Neither in the NisBC nor the ProcM enzymatic

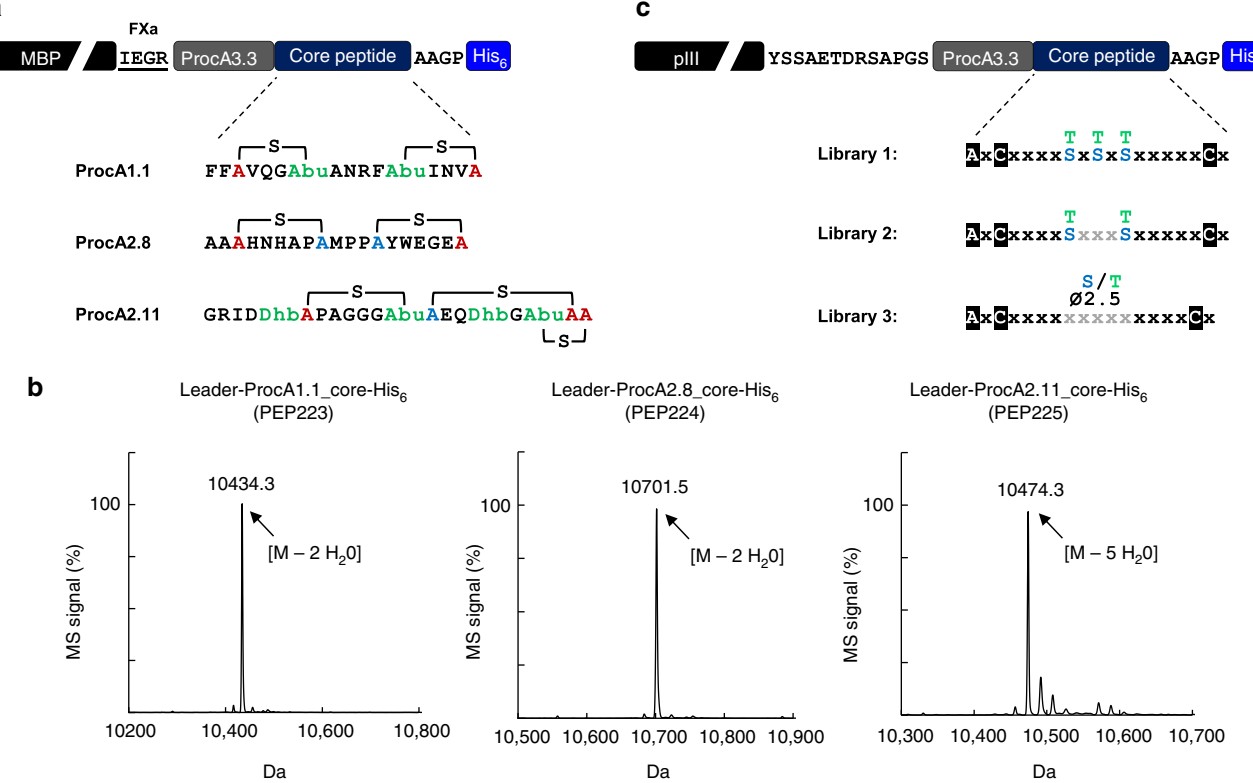

**Fig. 3** ProcM-mediated enzymatic modification of natural prochlorosin variants fused to the C-terminus of MBP and design of phage display libraries. **a** Sequence of natural prochlorosin core peptides with thioether bridges (brackets) and modified residues (colored) in context of C-terminal MBP fusions (the ProcA3.3. leader, linker sequences with FXa site and His₆-tag are indicated; for full sequences see Supplementary Table 5). **b** ESI mass spectra of prochlorosins produced as C-terminal MBP fusions in presence of ProcM co-expression and analyzed as leader-core peptides after proteolytic removal of MBP and IMAC purification. The main peak peptide masses reflect fully dehydrated core peptides and the number of identified dehydrations is indicated (each −18 Da). **c** Schematic representation of lanthipeptide precursor libraries fused to the C-terminus of pIII (linker sequences as indicated). In the library sequences white, boxed letters indicate fixed residues, x (black) represents all possible amino acids (except cysteine, threonine, and serine) at equal distribution. In library design 1 and 2, three or two fixed threonines or serines (blue and green) were incorporated at equal distribution, respectively. X (gray) in library 2 indicates all possible amino acids (except cysteine) at equal distribution. In library design 3, serines or threonines were distributed in a central stretch (indicated by x; gray) at an average rate of 2.5 per molecule with all other amino acids (except for cysteine) equally allowed

system FXa cleavage-resistant peptides were observed, which suggests that the displayed peptides are largely linear and failed to be efficiently modified (Fig. 2c, d, left panels). In contrast to the widely used N-terminal fusions to phage pIII, an isolated report describes fusions to its C-terminus. The C-terminus of pIII is directed toward the phage core and it was believed that fusions to the C-termini would not be easily accessible in display systems. However, Fuh and Sidhu[24] selected His₆-tagged randomized linker libraries fused to the C-terminus of pIII for binding to anti-His antibodies and identified 10 residue linker sequences that enabled display on the pIII C-terminus at similar rates as compared to conventional N-terminal display. We therefore established C-terminal pIII fusions of the same NisA- and ProcA-leader containing reporter peptides and assessed their modification status on phage as described above. In this display mode, a significant fraction of FXa cleavage-resistant peptides was observed on phage produced upon co-expression of both the NisBC (Fig. 2c, right panel) and the ProcM (Fig. 2d, right panel) enzymes. Mutating the serine or cysteine residues required for thioether formation to alanine abrogated FXa resistance. Moreover, when the size of phage-displayed monocycles was gradually increased by insertion of additional residues between the serine and cysteine residues, highly significant peptide modification was observed for all tested constructs (Supplementary Fig. 1). These results demonstrate that C-terminal lanthipeptide fusions to pIII

are recognized and modified by the class I and class II biosynthetic machinery, are further incorporated into phage particles, and are displayed in a solvent accessible manner.

**Design and selection of lanthipeptide phage display libraries**. Due to its unprecedented substrate tolerance, the class II ProcM enzymatic system[21] seemed to be the ideal starting point for the generation of lanthipeptide phage libraries displayed on the pIII C-terminus. A comparison of natural prochlorosin peptides with confirmed structure revealed an interesting feature that is present in several of the variants and might serve as a scaffold for library design. Even though unrelated in the primary sequence, ProcA1.1, 2.8, and 2.11 have two larger non-overlapping thioether bridges in common, that are formed from central dehydrated positions to cysteines located further outside in opposite orientation (Fig. 3a). To test whether these natural peptide substrates would be efficiently modified by the ProcM enzyme when expressed in the context of a large fusion protein, we used the maltose-binding protein (MBP) as carrier and surrogate. MBP has approximately the same size as phage pIII (~43 kDa) and comes with the advantage of soluble expression instead of being inserted into the *E. coli* inner membrane, which facilitates more detailed peptide analytics by electrospray ionization mass spectrometry (ESI-MS). We produced these peptides as C-

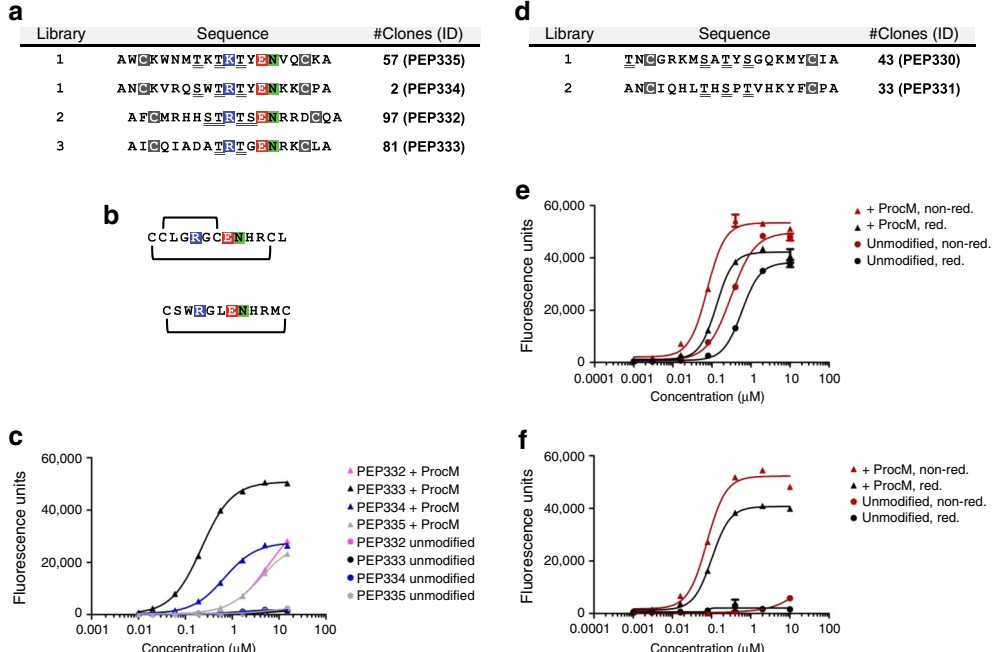

**Fig. 4** Phage selection outcome and characterization of identified lanthipeptides. **a** Amino-acid sequences of specific clones selected on uPA. The library in which the clones were found, the number of clones with identical sequences, and a unique clone ID are indicated. Fixed cysteine positions are boxed (gray), putative sites of dehydration (serines and threonines) are underlined, and a conserved three residue motif is highlighted (colored). **b** Sequence and disulfide pattern of bicyclic and monocyclic uPA-specific peptides described in the literature. **c** ELISA for uPA binding of purified His$_6$-tagged leader-core peptides produced with (+ProcM) or without (unmodified) ProcM co-expression. Data representing mean ± s.d. of three replicates is shown (curve fit with nonlinear regression). **d** As in **a**, but amino-acid sequences of specific clones selected on streptavidin are shown. **e** ELISA for streptavidin binding of purified His$_6$-tagged leader-core peptide PEP330 produced with or without ProcM co-expression and analyzed under reducing (red) and non-reducing (non-red.) conditions. **f** As in **e**, but binding curves of PEP331 are shown. Experiments shown in **c**, **e**, and **f** were repeated three times

terminal His$_6$-tagged fusions to MBP in presence of ProcM, removed MBP by proteolytic cleavage, and analyzed the modification status of purified leader/core peptides by ESI-MS. All three peptides were fully dehydrated by ProcM in the fusion context (Fig. 3b) and the formation of the correct bicyclic core in ProcA2.8 was further confirmed by electron transfer dissociation (ETD) fragmentation (Supplementary Fig. 2; Supplementary Tables 6–8). To further demonstrate that even more complex wild-type thioether cycle topologies are correctly introduced in the context of precursor peptides fused to the C-terminus of MBP, we expressed the lantibiotics nisin and lacticin 481 as MBP fusions and confirmed their antimicrobial activity (Supplementary Fig. 3). Encouraged by these results, three related lanthipeptide libraries reminiscent of the ProcA1.1/2.8/2.11 architecture (Fig. 3c) were designed and cloned as C-terminal pIII fusions, which resulted in >1 × 10$^9$ independent transformants for each library. Phage pools produced with ProcM co-expression were then selected in three iterative rounds of biopanning for binding to uPA (libraries 1, 2, and 3) or streptavidin-coated magnetic beads (libraries 1 and 2). After the third round of selection, the peptide encoding sequences of the enriched phage pools were subcloned as C-terminal fusions to MBP, co-expressed with ProcM in 384 well format, and cell lysates were screened for target binding by ELISA. Screening of the three libraries selected on uPA resulted in 79, 117, and 81 hits, out of 368 tested clones, respectively, and hit sequencing revealed the enrichment of certain clones (Fig. 4a). The enriched sequences from the different libraries share a common motif with highly conserved tandem glutamic acid, asparagine residues, and a positively charged residue at the −3 position. Strikingly, this motif has also been previously identified in specific uPA-inhibiting peptides selected from mono- or bicyclic disulfide bridged phage display

libraries[25, 26] (Fig. 4b). We produced and purified the selected variants as leader-core peptides with or without ProcM co-expression, assessed their modification status by ESI-MS, and tested their binding properties to uPA in an ELISA. All peptides produced with ProcM co-expression contained substantial amounts of fully dehydrated species (Supplementary Table 9) and showed specific binding to uPA, whereas no binding was observed when ProcM was omitted during expression (Fig. 4c). Furthermore the selected peptides inhibited the catalytic activity of uPA according to their binding profile and in a ProcM-dependent manner (Supplementary Fig. 4). More detailed analysis revealed that both, the ProcM-modified PEP332 and PEP334 contained two lanthionines as the dominant species among minor side products, whereas the major product in PEP333 and PEP335 rather contained a single lanthionine (Supplementary Fig. 5; Supplementary Tables 9–12). Even though we did not fully characterize all active species at this stage of the study, our results clearly show that post-translational modification of the selected core peptides by ProcM is a strict requirement to adopt structures that support uPA binding and inhibition.

Screening of the two libraries selected on streptavidin beads resulted in the identification of 49 and 33 hits, which represented two unrelated sequences (Fig. 4d), one of them having acquired an alanine to threonine mutation in the first position of the core peptide. Leader-core peptides produced in absence or presence of ProcM expression were analyzed and tested for streptavidin binding under reducing and non-reducing conditions. While PEP331 was fully dehydrated, one position in PEP330 had largely escaped dehydration (Supplementary Table 9). PEP330 bound streptavidin regardless of ProcM co-expression and reducing conditions had only minor effects, which suggests binding of a linear epitope (Fig. 4e). In contrast, the binding of PEP331 to

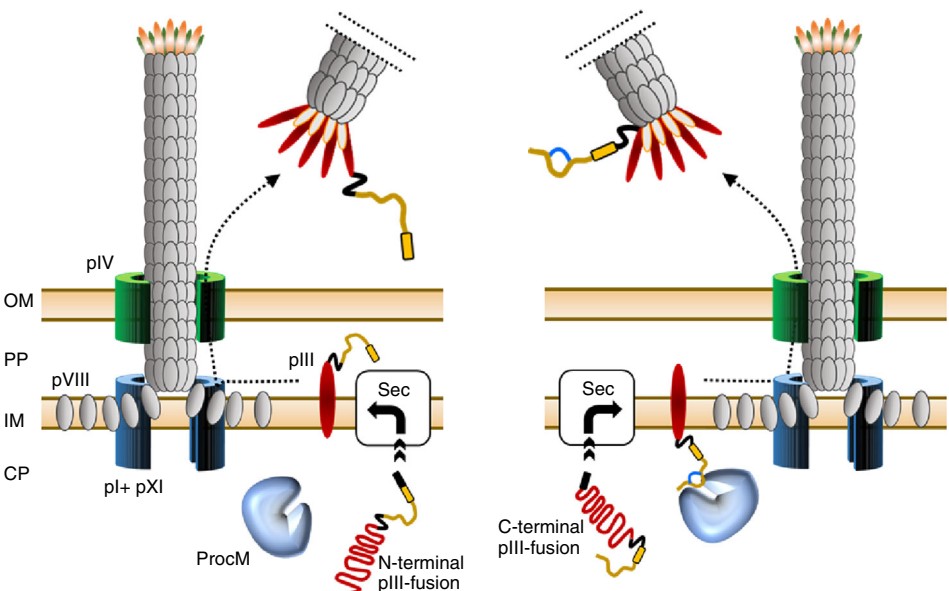

**Fig. 5** Display of lanthipeptide precursors on the N- and C-termini of phage pIII. N-terminal pIII display of largely unmodified lanthipeptide precursors (left panel). Translocation of unstructured peptide–pIII fusions from the cytoplasm (CP) via the Sec pathway (Sec) is fast allowing little or no lanthionine introduction by ProcM, whereas modified lanthipeptides are poorly transported via the narrow Sec pore. Prior to phage assembly, the precursor peptide (ocher) is exposed to the periplasm (PP) and no longer accessible to ProcM. C-terminal display of lanthipeptides (right panel). After translocation of the pIII–peptide fusion, the C-terminal exposed peptide remains in the cytoplasm allowing ProcM-catalyzed lanthionine introduction (indicated by a blue cycle). Phage display of modified peptide is accomplished after incorporation of pIII into the phage coat and subsequent extrusion of the phage particle into the medium. Black arrows indicate movement of capsid proteins to the phage assembly site. OmpA signal sequence in pIII fusions (black box), leader peptide (yellow box), linker sequences (black line), relevant phage coat proteins (as numbered), outer membrane (OM), and inner membrane (IM) are highlighted

streptavidin was strictly dependent on post-translational modifications introduced by ProcM and only a slight reduction in binding was noticed in a reducing environment (Fig. 4f). In-depth analysis confirmed the presence of two efficiently formed lanthionines in both PEP330 and PEP331 that resemble the structural topology found in native ProcA2.8 and which inspired the design of the library scaffold (Supplementary Fig. 5; Supplementary Tables 9, 13–15).

## Discussion

In summary, we show that the de novo selection of functional lanthipeptides by phage display can be achieved by fusing peptide precursor libraries containing natural leader sequences and a randomized core to the C-terminus of pIII. The co-expression of the lanthipeptide synthetase ProcM along with pIII-peptide precursor fusions leads to the sequential post-translational dehydration and cyclization of the core peptides in the cytoplasm of the producing cells and to their subsequent display on phage particles. We provide two examples in which biopanning of lanthipeptide display libraries resulted in the identification of peptide ligands that depend on ProcM-mediated post-translational modification for recognition of their targets, namely uPA and streptavidin. Interestingly, lanthipeptides selected on uPA share a known sequence motif with mono- and bicyclic disulfide-bridged peptide ligands previously identified by phage display. Moreover, the structures of these disulfide bridged peptides in complex with the catalytic domain of human uPA have recently been solved and their cyclic nature was reported to be indispensable for target binding[25–27]. The library design in this study was based on the thioether configurations described for some natural prochlorosins and aimed at providing a subtle excess of serines/threonines at flexible positions (to be enzymatically dehydrated) over fixed cysteines within the core peptide to increase the chance of subsequent cyclization. Even though five of the six characterized lanthipeptides identified in this study clearly

show the intended lanthionine configuration of two non-overlapping cycles, our analysis revealed that excess of available serines and threonines not necessarily correlates with improved cyclization efficiency, but might rather lead to stable phosphorylated intermediates, which fail to undergo phosphate elimination and conversion to the dehydrated residues[28]. In addition, some of the peptide preparations contained multiple modification products, which might complicate the identification of the active species (Supplementary Fig. 6, Supplementary Table 9). While site-specific peptide phosphorylation might be of interest for certain applications[29], this should be taken into account to further improve future library designs and emphasizes that the substrate requirements of promiscuous LanM enzymes are still not fully understood.

Fusions to the C-terminus of pIII are rather unconventional and to our knowledge have not been reported in the literature anymore since the first proof-of-concept study published by Fuh and Sidhu[24]. As emphasized by the authors, C-terminal fusions should be advantageous for the display of cDNA libraries and proteins that require a free C-terminus to support protein–protein interaction[24]. However, as demonstrated in this study, C-terminal fusions to pIII are furthermore ideally suited for the display of peptides that need to undergo post-translational modification in the cytoplasm of the host cell prior to phage assembly. We employed the widely used *ompA* signal sequence to target pIII-peptide fusions to the secretory (Sec) pathway. Upon translation, pIII is rapidly translocated and spans the bacterial inner membrane with the N-terminus facing the periplasmic space and the C-terminus remaining in the cytoplasm[30]. Hence, the fusion of lanthipeptide precursors to the C-terminus of pIII ensures prolonged exposure to the cognate modifying Lan enzymes located in the cytoplasm prior to phage assembly and further enables the display of cyclic peptides on phage particles (see Fig. 5 for detailed illustration). Besides this spatiotemporal advantage, the translocation of modified lanthipeptides fused to

the C-terminus of pIII is unlikely to be restricted by the pore diameter of the Sec translocon. The SecYEG channel is limited to the transport of proteins <2.9 nm in diameter[31], which would preclude the translocation of bulky cyclic peptides, such as nisin with estimated dimensions of 2.2 × 2.7 × 4.2 nm[32], fused to the N-terminus of pIII. This assumption is further supported by the failure to target nisin to the Sec pathway in the natural producer *L. lactis*[33]. In C-terminal display, the lanthipeptide fusion bypasses the SecYEG channel and only has to be threaded through the larger phage pIV pore (6–8.8 nm[30]) during phage extrusion. However, all of the five phage coat proteins (pIII, pVI, pVII, pVIII, and pIX) have been exploited for phage display[34] and we cannot rule out at this point that other display modes might also be suitable for the engineering of lanthipeptides. C-terminal peptide display has further been demonstrated on phage pVI[35] and pVIII[36], even though the reported display rates are well below the levels achieved on the pIII C-terminus, which we estimated to be ~0.3 per phage for lanthipeptide precursors. The Sec pathway translocates proteins in their unfolded conformation, whereas the twin-arginine translocation (Tat) pathway catalyzes the translocation of secretory proteins that fold in the cytoplasm[37]. It is intriguing to speculate that the Tat pathway might present an alternative to achieve prolonged lanthipeptide precursor interaction with Lan enzymes in the cytoplasm and to further support the efficient display of lanthipeptides on the N-terminus of pIII, but it should be noted that our preliminary attempts were not successful. The bioactivity of several Lan enzymes has been reconstituted in vitro and it seems plausible to modify linear lanthipeptide precursors displayed on phage or on ribosomes by incubation with recombinant Lan enzymes. However, the production of Lan enzymes is laborious, cysteines in the precursor peptides have to be trapped in a reduced state to allow for the enzymatic introduction of the thioether cycles, and the buffers and additives used during the reaction have to be strictly controlled[38]. In contrast, the here reported phage display platform essentially follows standard protocols and should be widely accessible to other laboratories.

During the establishment of the phage display procedure, we employed a modular two-plasmid system for the co-expression of Lan enzymes and lanthipeptide precursors that allowed to easily swap and test genetic parts, such as promoters, plasmid replicons, or enzyme- and peptide-encoding genes. We found that expression of ProcM from a low-copy plasmid with only ~3 copies per producer cell and under control of an inducible sigma 70-dependent (housekeeping) promoter is sufficient to display and select post-translationally modified lanthipeptides on phage. This suggests that the system could be further simplified by integration of the *procM* gene into the *E. coli* chromosome, thereby eliminating the need for a second plasmid. It was previously known that Lan enzymes recognize and modify lanthipeptide precursors with short N-terminal extensions, e.g., affinity tags, but the finding that also large bulky protein fusions, such as MBP and pIII, are tolerated was not anticipated. Beyond the instrumental impact for successful phage display on the pIII C-terminus, MBP fusions to lanthipeptide precursors warrant further attention. MBP is an excellent carrier protein that confers high level and soluble expression of various fusion partners that would otherwise suffer from poor producibility[39]. This applies also to lanthipeptide precursor fusions and enabled not only high-level soluble expression in *E. coli*, but also affinity screening of lanthipeptides in cell lysates in a 384-well format.

We believe that the here described phage display system is widely applicable to generate lanthipeptide ligands to protein targets of choice and can be adapted to other library designs, the use of alternative promiscuous lanthipeptide synthetases[8], or to incorporate further PTMs, such as glycosylations[40] or D amino

acids[41]. Moreover, the underexplored C-terminal display on pIII should be suitable for the engineering of other RiPP classes that rely on leader peptide-dependent enzymatic modification and could be a valuable source to identify therapeutic peptide candidates.

## Methods

**Overview of expression plasmids and phagemids.** The vector maps of the plasmids and phagemids used in this study and their essential features are shown in Supplementary Fig. 7. A detailed description of the plasmids is provided below. A list of oligonucleotide primers used for PCR and cloning is provided in Supplementary Table 16.

**Cloning of Lan enzyme expression plasmids.** All Lan enzyme encoding plasmids for expression in *E. coli* MC1061F′ were derived from pZE12MCS (Expressys). The ColE1 origin was swapped for RSF1030 by *Avr*II/*Spe*I cloning of a PCR product obtained on pRSFDuet-1 (Novagen) using primers JUB-001 and JUB-002 resulting in the intermediate plasmid pZE_RSF_MCS. The *nisC* and *nisB* genes from pRSFDuet-nisC-nisB4, encoding the chromosomal sequences from *L. lactis* NZ9700 under the control of two independent T7 promoters, were independently PCR amplified using primer pairs JUB-012/-013 and JUB-014/-015, respectively, and fused by a second step PCR using primers JUB-015/-16. *Kpn*I/*Mlu*I cloning of the product into pZE_RSF_MCS resulted in pZE_RSF_nisCB that encodes a bicistronic operon with N-terminally *myc*- or HA-epitope-tagged *nisC* and *nisB*, respectively. The *procM* gene was PCR amplified from chromosomal *Prochlorococcus* MIT9313 DNA (Provasoli-Guillard National Center for Marine Algae and Microbiota) using primers JUB-007/-009, digested *Kpn*I/*Asc*I, and ligated into *Kpn*I/*Mlu*I digested pZE_RSF_MCS. The RSF1030 origin in this intermediate was swapped for pSC101* by *Avr*II/*Spe*I cloning of a PCR product generated on pZS*13luc (Expressys) using primers JUB-136/-137 to obtain plasmid pZE_101_-procM. For T7-polymerase-driven expression, *procM* from pZE_101_procM was PCR amplified using primers JUB-071/-072 and cloned via *Nde*I/*Kpn*I into pRSFDuet-1 resulting in plasmid pRSFD_procM.

**Phagemid cloning for lanthipeptide display.** The phagemid pL3_stuffer that was used for display of precursor peptides on the N-terminus of pIII was obtained via cloning intermediates that were designed for peptide display on the major coat protein g8p. In a first step a synthetic DNA fragment (GeneArt) consisting of tandem *Xho*I/*Not*I sites followed by a His6-tag, a GGGDSRGGGAAGGGCDSRG GG linker with *Kas*I-site, the g8p coding sequence, two STOP codons and a *Hin*dIII site was cloned via *Xho*I/*Hin*dIII into pMORPH18[42]. In a second cloning step, the Shine-Dalgarno (SD) and *ompA* signal sequences from pMORPH30[43] were PCR amplified using primers JUB-023/-024 and introduced via *Xba*I/*Bam*HI into the aforementioned intermediate. In a final step, a truncated version of bacteriophage fd pIII (CT domain) was PCR amplified from pMORPH30 using JUB-019/-020 and cloned *Kas*I/*Hin*dIII to replace g8p, which resulted in the final plasmid pL3_stuffer. Synthetic DNA fragments encoding precursor lanthipeptides for display on the pIII N-terminus were cloned into the *Bam*HI/*Not*I sites of pL3_stuffer. The phagemid pL3C_zeoR-stuffer that was used for display of precursor peptides on the C-terminus of pIII was constructed by cloning of a synthetic DNA fragment encoding a SD sequence preceded by an *Xba*I site and a translational fusion consisting of an *ompA* signal sequence, the bacteriophage fd pIII CT domain, a linker sequence (YSSAETDRSAP[24]), a zeocin resistance gene (*ble*) flanked by *Bam*HI and *Hin*dIII restriction sites, via *Xba*I/*Hin*dIII into pL3_stuffer. A *Sal*I restriction site was introduced downstream of the *Hin*dIII site by PCR using primers JUB-230/-231 to obtain the final plasmid. All C-terminal pIII-peptide fusions were established from double-stranded synthetic DNA fragments (GeneArt) and cloned into the *Bam*HI/*Hin*dIII or *Nde*I/*Hin*dIII sites of pL3C_zeoR-stuffer. The resulting amino acid sequences of all pIII-peptide fusions are shown in Supplementary Tables 3, 4. DNA fragments encoding lanthipeptide display libraries were inserted into the *Xba*I/*Sal*I sites of pL3C_zeoR-stuffer as described below.

**Cloning of reporter lanthipeptides for expression in *E. coli* MC1061F′.** All expression constructs for reporter lanthipeptides were established from double-stranded synthetic DNA fragments (GeneArt) and cloned into the *Xba*I/*Hin*dIII sites of plasmid pL3_stuffer, which replaced the phage pIII gene by the coding sequence of the precursor peptide. The resulting amino-acid sequences of precursor peptides are shown in Supplementary Tables 1, 2.

**Cloning of lanthipeptides for production in *E. coli* BL21(DE3).** The pET21a_MBP_FXa_zeoR-stuffer plasmid used for T7-polymerase-driven production of lanthipeptides as C-terminal fusions to MBP was constructed by *Xba*I/*Hin*dIII cloning of a synthetic double-stranded DNA fragment (GeneArt) encoding the *malE* gene (coding residues 27–391 preceded by an ATG start codon) fused to a linker sequence with integrated FXa cleavage site (sequence NSSSNNNNNNNNN NGTIEGRGSPGGSGGAPGS; FXa site underlined) and a zeocin resistance gene (*ble*) flanked by *Bam*HI/*Hin*dIII sites into pET21a (Novagen). All prepeptide

sequences either derived from synthetic DNA fragments or PCR products obtained on the phagemid libraries using the primer pair JUB-225/-248 were cloned into the BamHI/HindIII sites of pET21a_MBP_FXa_zeoR-stuffer. The amino-acid sequences of the resulting precursor peptide fusions are shown in Supplementary Table 5.

**Library syntheses.** The DNA fragments of the variable regions of the lanthipeptide libraries were synthesized by a modified version of the Slonomics Technology for de novo gene synthesis[44], which facilitates both, the consecutive generation of variable positions, as well as the synthesis of a constant sequence by parallel incorporation of complex anchor mixtures during elongation. Herein, every ligation-mediated elongation step is followed by a specific cleavage with a type IIS restriction enzyme generating new overhangs for a further reaction cycle in which a new mixture of anchor molecules (or an individual anchor) is added to the growing DNA fragment through subsequent ligation reactions. Intermediate reaction products were separated from undesired by-products through repeated immobilization of biotinylated DNA fragments on a streptavidin-coated surface during every cycle of the synthesis process. Following the outlined procedure, any variable sequence motif at any position of a given DNA sequence could be introduced, while allowing the precise control of the type, number, and frequency of individual codons. To build the variable DNA fragments of the lanthipeptide display libraries, mixtures of anchor molecules with defined ratios were prepared according to the amino-acid distribution specifications of the DNA library designs (Fig. 3c). Each individual design was converted into a list of anchor mixtures and single anchor molecules that was used as a template for automated pipetting by a robotic platform to generate the highly complex molecule mixtures used in the synthesis procedure. The syntheses of the diverse DNA sequence fragments were carried out by repeated cycles of ligations of these defined anchor mixtures to precursor molecules and subsequent restriction reactions, as described previously[45]. Variable sub-fragments were assembled using a type IIS restriction endonuclease-based restriction digest and ligation procedure that allowed the seamless joining of the randomized DNA fragments. Upon completion of the assembly procedure, by joining variable fragments with constant flanking regions, the final DNA library fragments were cloned (via XbaI/SalI) into the lanthipeptide display vector pL3C_zeoR_stuffer and quality control was performed by analyzing an adequate number of clones by Sanger sequencing.

**Expression and purification of ProcM-modified lanthipeptides.** Selected lanthipeptide candidates were co-expressed as C-terminal fusion to the MBP from pET21a_MBP_FXa_zeoR-stuffer along with ProcM encoded on pRSFD_procM in E. coli BL21 (DE3) cells (Novagen). 500 ml 2× YT cultures containing 100 μg ml$^{-1}$ ampicillin and 100 μg ml$^{-1}$ kanamycin were inoculated from 4 ml overnight precultures and incubated at 22 °C, 230 rpm until an $OD_{600}$ of 0.4–0.6. Expression was induced by the addition of IPTG to a final concentration of 0.75 mM and cultures were further incubated for 20–22 h at 22 °C, 230 rpm. All expressions were performed in 1 l centrifuge beakers (Heraeus). Cells were harvested by centrifugation at 6000×g for 30 min at 4 °C (Cryofuge 8500i Heraeus; Rotor 6606/8165). Then the pellet was frozen at −20 °C for 2 h or until further processing. The bacterial pellets were resuspended in 20 ml lysis buffer (phosphate-buffered saline (PBS) supplemented with 2 mM $MgCl_2$, 20 U ml$^{-1}$ Benzonase (Roche) and one tablet per 50 ml complete, EDTA-free protease inhibitor cocktail tablets (Roche)) by shaking at 200 rpm for 30 min at room temperature. Cells were disrupted either by chemical lysis (addition of 0.2% (w v$^{-1}$) lysozyme (Roche) to the lysis buffer) or high-pressure homogenization (three passes at 1 Mbar using the EmulsiFlex-C3 high-pressure homogenizer (Avestin)). The resulting suspension was centrifuged at 16,000×g for 30 min at 4 °C and the supernatant was sterile filtered (Millipore steriflip filters, pore size 0.22 μm) for further purification steps. The MBP–lanthipeptide fusions were purified by Dextrin–Sepharose affinity chromatography using 1 or 5 ml MBP-Trap columns (GE Healthcare) or self-packed Dextrin–Sepharose (GE Healthcare) 1 ml columns. The proteins were eluted from the columns with PBS containing 10 mM maltose. The purified MBP fusions were buffer-exchanged by PD10 columns (GE Healthcare) in FXa-digest buffer (20 mM Tris/HCl pH 8.0, 100 mM NaCl, 2 mM $CaCl_2$) and the peptides were released from MBP by addition of FXa (1:100 (w w$^{-1}$), NEB) and incubated overnight in a rotary shaker at 22 °C. The released peptides were further purified by IMAC using either 1 ml Protino IMAC columns or self-packed 1 ml columns filled with Profinity IMAC Ni-charged resin. All affinity chromatography steps were performed using the ÄKTA Avant 25 preparative chromatography system for low-throughput applications and using the Gilson ASPEC GX-274 solid-phase extraction workstation for high-throughput applications. Buffer exchange to 10 mM Na–phosphate buffer, pH 7.4, was performed using PD 10 columns. Samples were sterile filtered (0.2 μm pore size) and peptide concentrations were determined by UV spectrophotometry (Nanodrop, Thermo Fisher). The purity of the samples was analyzed by denaturing, reducing, or non-reducing 15% sodium dodecyl sulfate–polyacrylamide gel electrophoresis (Bio-Rad) and by RP-UPLC.

**Factor Xa cleavage reporter assay.** Plasmids encoding reporter precursor peptides with an FXa site in the core sequence (either as prepeptides alone or translationally fused to phage pIII) were transformed into chemical competent E. coli

MC1061F′ established with or without a second plasmid encoding the cognate Lan enzyme(s) in parallel. Glycerol stocks of three independent transformants were stored at −80 °C in 96 well plates. For reporter assays on solubly expressed peptides, biological triplicate precultures were inoculated from glycerol stocks in 96 well plates containing 2xYT medium with the appropriate antibiotics and grown overnight at 37 °C, shaking. About 24 well cultures (2xYT medium with 0.1% glucose and antibiotics) were inoculated from precultures, grown at 37 °C until early logarithmic phase, and cells collected by centrifugation. The cells were resuspended in 2xYT induction media containing 0.5 mM IPTG to allow peptide and Lan enzyme expression to proceed for 20 h at 22 °C. Cells were collected and lysed (in 1× BBS, 2.5 mg ml$^{-1}$ lysozyme, 50 U ml$^{-1}$ Benzonase, 1 mM EDTA) for 1 h at 22 °C, 500 rpm. Lysates were cleared by centrifugation and added in quadruplicate to 384 well MA2400 plates (Meso Scale Diagnostics) coated with anti-His tag IgG (R&D Systems, MAB050; 3 μg ml$^{-1}$). After affinity capture of the peptides for 1 h, plates were washed six times with TBS and once with FXa reaction buffer (20 mM Tris-HCl, 50 mM NaCl, 1 mM $CaCl_2$, pH 6.5). Two of the quadruplicate samples were treated with 500 nM FXa for 20 h at 22 °C, the others incubated with FXa reaction buffer alone (technical duplicates). After removing FXa-digested peptide fragments by washing five times with PBST, intact peptides were detected with either biotinylated anti-FLAG (Sigma, # F-9291) or anti-HA antibodies (Thermo Scientific, #26183-BTIN) and MSD GOLD SULFO-TAG (Meso Scale Diagnostics) conjugated streptavidin. Electrochemiluminescence was measured on a Meso Scale Discovery SECTOR Imager 6000. Signals obtained without FXa treatment were set to 100% (input) and the signal ratio for the corresponding sample after FXa treatment was calculated (signal remaining after FXa treatment (%)). Obtained values therefore reflect the percentage of FXa cleavage-resistant peptide and are a direct measure of thioether bridge formation efficacy. For FXa reporter assays on phage displayed precursor peptides, triplicate glycerol stocks containing E. coli strains harboring phagemids with or without the cognate Lan enzyme plasmid were grown overnight in 96 well plates containing media supplemented with the appropriate antibiotics. The following day 10 μl of these saturated cultures were used to inoculate 24 well plates containing 750 μl fresh media per well, and cultures grown at 37 °C, shaking. At early logarithmic phase, cultures were infected with VCSM13 helper phage at a multiplicity of infection of ~10. Cells were collected by centrifugation, resuspended in 3 ml 2xYT induction media containing 0.5 mM IPTG and appropriate antibiotics, and phage production allowed to proceed for 20 h at 22 °C. Phage supernatants were cleared of producer cells by centrifugation and transferred in quadruplicate to 384 well MA2400 plates coated with anti-M13 IgG (GE Healthcare, #27-9420-01; 3 μg ml$^{-1}$). FXa treatment, detection of the displayed peptides, and data analysis were performed as described above.

**Statistical analysis.** All statistical analyses were performed by using GraphPad Prism (version 5.04, GraphPad software Inc., San Diego, CA, USA). A two-sided, unpaired Student's t-test was used to analyze differences between means. Data are shown as mean ± s.d. A p-value <0.05 was considered statistically significant.

**Phage selection on uPA and streptavidin-coated magnetic beads.** The number of transducing units (t.u.) present in the established phage libraries was determined by infectivity titration on susceptible E. coli TG1F+ cells. Phage titers were in the range of 10$^{14}$ t.u. per ml (>10-fold library size) and no sign of reduced infectivity caused by C-terminally displayed peptides was noted. For the first round of selection, all phage libraries were blocked overnight at 4 °C in 500 μl Chemi-BLOCKER (Millipore) on a rotator. For pre-adsorption, phages were diluted to ~1.5 × 10$^{13}$ t.u. in 500 μl ChemiBLOCKER containing 0.05% (v v$^{-1}$) Tween 20, incubated for 30 min at room temperature twice with 1 mg streptavidin beads (Dynabeads M-280, ThermoFisher), and once with 2 mg of a randomly biotinylated Fab-fragment immobilized on 2 mg streptavidin beads. Pre-adsorbed phages were transferred to fresh 2 ml tubes, random biotinylated uPA (ProsPec, #enz-264-c; purified by size exclusion chromatography) added to a final concentration of 100 nM, and samples incubated 1 h at room temperature on a rotator. The samples were then captured on 2 mg streptavidin beads for 20 min at room temperature, washed five times each with PBS containing 0.05% (v v$^{-1}$) Tween 20 or PBS alone, and phages eluted by incubation with 200 μl 100 mM Glycin–HCl, 0.5 M NaCl, pH 2.2 for 10 min. After neutralization with 2 M Tris pH 8, phages were incubated for 45 min at 37 °C with 15 ml E. coli TG1 F+ harboring plasmid pZE_101_procM that were grown to an optical density at 600 nm ($OD_{600}$) of 0.5. The infected cells were harvested by centrifugation, plated on large LB/chloramphenicol/ampicillin plates, incubated over night at 37 °C, and used to produce phage for the next round of selection. Two additional selection rounds were performed with reduced uPA concentration (50 and 25 nM, respectively) and with elongated washing steps. The selection for streptavidin binding was performed essentially as described above using 1.2, 0.6, and 0.3 mg streptavidin-coated beads as target in round one, two, and three, respectively, and three pre-adsorption steps each on 1 mg Protein G Dynabeads (Life Technologies).

**Analysis of MBP-fused prochlorosin peptides by mass spectrometry.** Peptide samples were analyzed using an Acquity UPLC System (Waters) coupled to a Synapt G2 Si mass spectrometer with ETD capability (Waters). Peptide identity

was verified by intact mass measurement following analytical separation on an Acquity UPLC Protein BEH C4 column (2.1 mm × 50 mm, Waters) using a gradient of 3–60% acetonitrile in water over 20 min at a flow rate of 500 µl min⁻¹. Each eluent was supplemented with 0.1% formic acid. The eluent was passed into the UV detector and the electron spray ionization (ESI) source via a 15:1 splitter (Waters). The time course of elution was recorded by UV at 214 nm and by MS at a source voltage of 0.8 kV, simultaneously. MS data were processed using MaxEnt3 (Waters) for MW < 10 kDa and MaxEnt1 (Waters) for MS > 10 kDa. The results of all ESI-MS analyses are summarized in Supplementary Table 6. ETD mass spectrometry was used to analyze the structure of leader free ProcA2.8. Following analytical separation on an Acquity UPLC BEH C8 column (2.1 mm × 100 mm, Waters, 3–60 % acetonitrile in water over 20 min, flow rate 150 µl min⁻¹), the eluent was passed directly into the ESI source. The 6+ charged ion ($mz^{-1}$ 538.92) of the peptide was selected as precursor and fragmented under ETD conditions using 4-nitrotoluene as an electron donor reagent. The ETD spectrum was analyzed by assigning the ion signals to the c- and the z fragments ($c^{1+}$, $c^{2+}$, $c^{3+}$, $z^{1+}$, $z^{2+}$, and $z^{3+}$) expected from the ProcA2.8 sequence. While fragments before, between, and after the proposed thioether bridges were found (Supplementary Fig. 2d; Supplementary Tables 7, 8), no fragments from the sequence within the proposed thioether bridges could be assigned indicating that the lanthionines have been formed in a consecutive arrangement.

**Subcloning and affinity screening of selected clones**. After the third round of selection plasmid DNA was isolated, the precursor peptide sequences PCR amplified using the 5′-biotinylated primers JUB-225/-248, parental plasmid DNA digested with DpnI, and the PCR products purified via silica membrane columns (Promega). The purified PCR products were digested with BamHI/HindIII and three times sequentially passed over streptavidin-coated 96 well plates (Microcoat) to capture released biotinylated overhangs and undigested DNA. Without further purification the inserts were ligated into the BamHI/HindIII digested vector pET21a_MBP_FXa_zeoR_stuffer and transformed into chemical competent E. coli BL21(DE3) harboring plasmid pRSFD_procM, and plated on large LB/ampicillin/kanamycin plates. 368 transformants from each library output were then picked into 384 well plates and stored as glycerol stocks at −80 °C. The glycerol stocks were used to inoculate fresh 384 well plates and grown overnight at 37 °C to obtain saturated cultures of similar density and viability. 5 µl of these starter cultures were then used to inoculate 50 µl 2xYT/ampicillin/kanamycin media in 384 well plates, the cultures grown until slightly turbid at 37 °C, shaking, and 10 µl media containing 3 mM IPTG (0.5 mM final) was added to induce expression of MBP-precursor peptide fusions and ProcM enzyme. After overnight growth at 22 °C, shaking, 15 µl lysis buffer (2× BBS containing 2.5 mg ml⁻¹ lysozyme, 4 mM EDTA, 50 U ml⁻¹ Benzonase) was added per well, and samples incubated for 2 h at room temperature. The lysates were blocked by adding 15 µl PBS containing 12.5% BSA and 0.3% (v v⁻¹) Tween 20 and stored at −20 °C until further use. To identify target specific clones from phage selections performed on uPA, 100 nM biotinylated uPA was captured on 384 well plates coated with NeutrAvidin (Pierce) overnight at 4 °C. The plates were washed three times with PBS containing 0.05% (v v⁻¹) Tween 20, blocked with 5% (w v⁻¹) BSA in PBS for 1 h at room temperature, and washed again as above. Cell lysates were added and peptide fusions to the C-terminus of MBP allowed to bind for 1 h at room temperature, shaking. After washing the plates as above, alkaline phosphatase conjugated anti-His antibody (Bio-Rad, #MCA1396A) was added, plates incubated 1 h, washed 5× with TBST, and fluorescence read in a Tecan M200pro after addition of AttoPhos substrate (Roche). In parallel, unspecific binding of the lysates was tested in the same way as described above using an equimolar amount of biotinylated Fab fragment instead of uPA captured on NeutrAvidin plates. The expression levels of MBP–peptide fusions in the cell lysates were assessed by capture on MaxiSorp plates (Thermo Scientific) coated with anti-His IgG (R&D Systems, #MAB050) followed by detection with rabbit polyclonal anti-MBP antibodies (Abcam, # ab9084) and alkaline phosphatase conjugated goat anti-rabbit IgG (Sigma, # A3687) according to the same procedure. To identify target specific clones from phage selections performed on streptavidin beads, cell lysates were tested in an ELISA as described above for binding to streptavidin (IBA) that was coated onto MaxiSorp plates at 200 nM overnight.

**ELISA using purified peptide**. Peptide candidates identified by phage display were produced and purified as leader-core peptides as described above. 384 well Maxisorp plates were coated with uPA or streptavidin (100 nM each) overnight at 4 °C. After washing and blocking the plates as described above, serial dilutions of the peptide preparations in PBST containing 0.5% (w v⁻¹) BSA (and 10 mM DTT for anti-streptavidin replicates) were added in duplicate, and plates incubated for 1 h at room temperature, shaking. Detection of bound peptides with AP-conjugated anti-His IgG was performed as described above.

**Synthetic peptides**. Peptides were chemically synthesized at JPT (Germany) using Fmoc (N-(9-fluorenyl)methoxycarbonyl) chemistry. The thioether-bridged peptide was produced by oxidation of cysteines followed by base-assisted desulfurization. Crude peptides were purified by employing C18 RP-HPLC and analyzed by ESI-MS. The lyophilized peptides were dissolved in $H_2O$ and stored at −80 °C until use.

**Data availability**. The authors declare that all the data supporting the findings of this study are available within the paper and its Supplementary Information files. Additional raw data are available from the corresponding author upon reasonable request.

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

## Acknowledgements

We thank Andrea Nudlbichler, Gertrud Hampe, Raphael Hechinger, Theresa Kober, and Jakob Schüle for excellent technical assistance and Marcel P. de Vries and Benjamin Hackner for support with mass spectrometry. We further thank Karin Felderer and Daniel Weinfurtner for fruitful discussion throughout the project.

## Author contributions

J.H.U. and M.A.M. conceived and designed experiments. T.A. and R.R. performed peptide analytics. K.G. and M.Z. performed most experiments. M.T. produced proteins and peptides. J.H.U., M.A.M., T.A., K.G., M.Z., R.R., G.N.M., and J.P. analyzed data. K.S. implemented and supervised library syntheses. T.B. provided idea for C-terminal display. K.T., G.N.M., and J.P. provided valuable input. J.P. supervised the project. J.H.U. wrote the manuscript with input from all authors.

## Additional information

**Competing interests:** J.H.U., M.A.M., J.P., and T.B. are inventors on a patent application related to the ideas described in this manuscript. The remaining authors declare no competing financial interests.

