## [Peer Review File · Nature Communications]

Reviewers' comments:

Reviewer #1, an expert in natural product biosynthesis (Remarks to the Author):

This manuscript details the development of a phage display assay for the selection of post-translationally modified peptides that bind to protein targets. The described experiments make two major advances in the field. First, they demonstrate that RiPP natural product biosynthetic enzymes (specifically, lanthipeptide synthetases) can be used to generate libraries of post-translationally modified peptides, and that these libraries can then be displayed on phage for selection experiments. This opens the door for using these enzymes to generate libraries of peptides with structures that are not easy to make with synthetic chemistry approaches. The relatively untapped structural space provided by these post-translationally modified peptides could enable the selection/evolution of peptides that bind to novel targets. Moreover, the fact that many of these RiPP peptides (including the lanthipeptides of the current study) contain enzymatically-installed macrocycles is of interest. The macrocycles of RiPPs often contribute greatly to their stability and target recognition; thus, they should serve as an excellent scaffold for the development of peptide-based biologic drugs. The second major advancement is the authors' realization that expressing the RiPP peptides to be modified on the C-terminus of the pIII coat protein is critical for their successful *in vivo* modification by the biosynthetic enzymes. Indeed, they showed that when the same peptides were expressed as the (more typical) N-terminal pIII fusions, no modification was observed. The authors speculate that the C-terminal fusions remain in the cytosol for longer periods of time, which may allow for more thorough and efficient modification by the co-expressed RiPP biosynthetic enzymes. A similar strategy could be useful for phage display of RiPPs other than lanthipeptides. Overall, the studies are highly meritorious and impactful and deserve publication in *Nat. Commun.* However, a few points need to be addressed prior to publication.

1. Could the authors provide some rationale for the choice of the SWXnIEGRXnEC model sequences shown in Figure S1? Was it necessary to include the Trp residue at the C-terminal side of Ser and the Glu residue at the N-terminal side of Cys? Were other model sequences tested? If so, a list of failed sequences provided in the S.I. would be useful for others considering using the NisB/C and/or ProcM enzyme systems for engineering applications.
2. Regarding the ProcM modification of MBP fusions (Figure 3 and S2). The MBP fusion strategy could be very useful for people interested in studying biosynthesis of RiPPs, as many RiPP precursor peptides have solubility issues. ETD data are only shown for ProcA2.8. Why aren't ETD data shown for 2.11 and 1.1? I'm curious if the topology of the modified 2.11 and 1.1 peptides are the expected wt topology (as shown in Fig. 3a), or whether the MBP fusion has perturbed it. It would be nice to see this data so that researchers contemplating the MBP fusion strategy will have a better feel for how often native topology is preserved.
3. Regarding the phage display/biopanning experiments of ProcM-modified libraries (Figure S3 and Figure 4). Authors show that ProcM co-expression is needed for peptides to bind uPA, clearly demonstrating that the peptides need to be post-translationally modified in order to bind target. However, no data on the cyclization state of the peptides is presented. It is implied that the peptides are cyclized by the comparison with previously generated S-S linked, phage displayed peptide libraries that bind to uPA, but the cyclization status of the peptides in the current study is not explicitly shown. The authors should report ETD data on the best-hit binders shown in Figure 4 to see if they are indeed cyclized. Cyclization will be critical for *in vivo* stability of any peptide biologic developed by this approach. Thus, it is highly important to demonstrate that ProcM is generating cyclized peptides in this assay.
4. Why is +/- ProcM data for PEP335 not shown in 4c?
5. Table S8. What is being reported? Average mass? This should be stated somewhere. Also, the

errors seem to be a little high (many are ~ 30 ppm or more). This instrument should be capable of single digit or less ppm errors. Could the authors comment on this?

Reviewer #2, an expert in cyclic peptides (Remarks to the Author):

In this manuscript, the authors describe the development of a phage display system suitable for the production and affinity selection of post-translationally modified lanthipeptides fused to the C-terminal domain of the minor pIII coat protein in M13 phage. The authors first examined and validated the production of fully processed (= thioether-linked) lanthipeptides in solution and fused to the phage using a clever protease-dependent ELISA assay (Fig. 1-2). These experiments yielded the key finding that C-terminal fusion of the precursor polypeptide to the pIII coat protein allows for post-translational maturation of the lanthipeptide (dehydration to some level of cyclization), whereas this was not possible using the standard N-terminal fusion approach. The proposed explanation for this difference, as described in Fig. 5, is reasonable. After demonstrating that the ProcM biosynthetic enzyme can produce a series of model prochlorosin lanthipeptides in the phage system (Fig. 3), the authors generated three large libraries of randomized precursor sequences (Fig. 3c) and panned the resulting peptide libraries against uPA or streptavidin. Enrichment and deconvolution of the libraries resulted in the identification of hit sequences that bind to each of these target proteins with hundred-nanomolar affinity (Fig. 4).

The methodology presented here is novel and, if fully validated (see below), potentially very impactful. Although the individual elements of the present system were known (e.g., C-terminal display on pIII, biosynthesis of lanthipeptides in *E. coli*), their successful integration for permitting the creation and screening of lanthipeptide libraries displayed on phage is an important accomplishment. That said, there are two major issues. First, the present results do not convincingly show that the hits isolated from the library screening against both targets are indeed cyclic, as proposed by the authors. This is a critical aspect since the relevance of this system is largely based on its capability of discovering new bioactive lanthipeptides. Second, none of the bioactive peptides isolated from the libraries was structurally characterized. Given the ambiguity concerning the position of the dehydrated amino acids and the presence, number, connectivity, and stereochemistry of the thioether linkages, demonstrating that unambiguous structural identification of the hits is possible becomes critical. Without that capability, the present would have little practical use. Other points that should be addressed during revision are discussed further below.

Major issues:

a) The authors should demonstrate the presence and, ideally, also the functional importance of the thioether linkage(s) in the sequence hits isolated against uPA and streptavidin (Fig. 4). The MS data of Fig S3 show the occurrence of Ser/Thr dehydration but no experimental evidence is provided to support the presence and number of thioether linkage(s) in these compounds. Several data actually argue against it. First, the data of Fig. 2 show that ProcM-catalyzed thioether formation in the presence of dehydrobutyrine is inefficient. Fig. 4e-f data show a decrease in streptavidin binding affinity for peptide PEP330 and PEP331 in the presence of reductant, suggesting that these peptides are cyclized via a disulfide bridge, not via a thioether linkage. The same experiment (+/- reductant) was not performed with the uPA binding peptides (Fig.4c), contributing to the uncertainty regarding the presence (and importance) of thioether linkages in these compounds.

b) The structure of at least one of the bioactive peptides isolated against each target should be unambiguously elucidated. Hit elucidation is far from being trivial in the present system. Take the strongest consensus sequence isolated against uPA (PEP332). Fig S3a shows that the major product carries three dehydrated S/T out of 4 possible ones. Considering all possibilities with

respect to the target AA for dehydration, plus all the possible thioether connectivities of a putative monocyclic and bicyclic product, there are >30 possible structures. This issue is further complicated by the fact that thioether formation with dehydroalanine (from dehydration of Ser) can occur even in the absence of enzymatic catalysis (as shown by data in Fig. 2a), thus generating two possible stereoisomers for each product containing such a bridge.

c) On the related note, it remains unclear if multiple products are actually formed after ProcM processing of the sequences described in Fig. 4a and 4d. The MS data of Fig S2 provide information only on the number of dehydrated amino acids, which may or may not be cyclized via a thioether bond. Thus, it is possible that multiple regio- and/or diastereomeric products could be formed in these reactions. HPLC or LC-MS traces of these reactions should be provided in the SI to clarify this point. This aspect should be commented upon in the text.

Other points:

a) The results of Fig. 2a-b show that the impact of S->T substitution on preventing ProcM-catalyzed cyclization is substantial. The same comparison should be made for the sequences fused to the C-terminus of the pIII coat protein (panels c and d of Fig. 2). This information is particularly important considering that all hit sequences contain one or more Thr within the target sequence.

b) The impact of lanthipeptide display on the infectivity of the phage particles should be measured and commented upon. While the authors provide information regarding the original size of the libraries, the chemical diversity that remains effectively accessible by the method is also dictated by how many phage particles can be propagated after panning. This information can be gained from simple control experiments.

c) The SI carries a section about peptide synthesis (page 21) but no synthetic peptides were discussed in the main text.

Reviewer #3, an expert in phage display (Remarks to the Author):

Lanthipeptides are enzymatically cyclized peptides with a thioether bridge that is resistant to reduction. The constraint reduces peptide flexibility and can confer higher affinity and specificity of binding to target proteins. Natural lanthipeptides have been found to exhibit antimicrobial and antiviral activities, and have thus attracted attention as potential drugs.

In this paper, the authors report a new method for the construction of large libraries of lanthipeptide synthesized and displayed on phage particles. To achieve this, they took advantage of a previously described C-terminal M13 pIII display system in which the C-terminally fused peptide resides in the bacterial host cytoplasm prior to phage assembly. By co-expressing the enzyme responsible for the formation of the lanthipeptide thioether bridge in the host bacteria, the authors show that significant quantities of cyclized lanthipeptides can be displayed on phage. Furthermore, the authors select random C-terminal lanthipeptide libraries against streptavidin and the serine protease urokinase plasminogen activator (uPA) and obtain binders in each case, and they show that the binding is dependent on incubation of the peptides with the cyclizing enzyme.

Overall, this is a well-written report of a clever method that should be of interest to many researchers. Moreover, the C-terminal phage display format could also be used to produce peptide libraries modified by other enzymes to further expand diversity beyond the genetically encoded amino acids.

However, the authors stop short of demonstrating that the peptides that they have derived exhibit activity. This could easily be tested with the peptides that bind to uPA, since this is a protease that can be assayed for activity in the presence or absence of the binding peptides to determine whether the peptides inhibit activity. It would be very worthwhile to determine the potency of the inhibitory activity of at least the highest affinity lanthipeptide for uPA, and to compare this to previous disulfide constrained peptides reported in the literature.

Point-by-point response to the reviewers' comments

Dear Reviewers,

Thank you very much for thoroughly reviewing of our manuscript. Thanks to your comments we have significantly improved our manuscript. Please find our answers to your comments below.

Reviewer #1

1. Could the authors provide some rationale for the choice of the SWXnIEGRXnEC model sequences shown in Figure S1? Was it necessary to include the Trp residue at the C-terminal side of Ser and the Glu residue at the N-terminal side of Cys? Were other model sequences tested? If so, a list of failed sequences provided in the S.I. would be useful for others considering using the NisB/C and/or ProcM enzyme systems for engineering applications.

Reply:

The SWXnIEGRXnEC model sequences shown in Figure S1 contains the same serine and cysteine flanking residues as the ones shown in Figure 2 in the main section. We established and screened a limited series of FactorXa cleavage site-containing model peptides and chose the serine and cysteine flanking residues based on previous work (Rink, R. et al. Lantibiotic structures as guidelines for the design of peptides that can be modified by lantibiotic enzymes. *Biochemistry* **44**, 8873–8882 (2005)). Even though the Trp and Glu positions were not essential and none of the tested residues completely abrogated thioether formation, the SWXnIEGRXnEC variant showed the highest degree of modification and was used for further studies. The limited set of tested sequences did not allow to draw general conclusions about the requirements for serine/cysteine flanking residues, but in combination with our FXa reporter assay this could be the scope of further studies.

The expression constructs used in this very early screen still encoded a free cysteine upstream of the leader peptide sequence and were initially designed for phage cys-display, in which a disulfide should be formed with a free cysteine on phage pIII in the periplasm of the producing cell. Since we decided not to follow up on this approach, we would rather omit to show the results in the manuscript to avoid an extra layer of complexity. However, the results of the screening are attached to this response letter as Appendix Figure 1.

2. Regarding the ProcM modification of MBP fusions (Figure 3 and S2). The MBP fusion strategy could be very useful for people interested in studying biosynthesis of RiPPs, as many RiPP precursor peptides have solubility issues. ETD data are only shown for ProcA2.8. Why aren't ETD data shown for 2.11 and 1.1? I'm curious if the topology of the modified 2.11 and 1.1 peptides are the expected wt topology (as shown in Fig. 3a), or whether the MBP fusion has perturbed it. It would be nice to see this data so that researchers contemplating the MBP fusion strategy will have a better feel for how often native topology is preserved.

Reply:

Our results show that leader recognition and efficient core peptide modification (dehydration) has occurred in all three tested variants and we singled out one variant for rather laborious ETD fragmentation that exemplifies correct ring formation and supports the library design in Figure 3C.

Due to the context (same MBP-Peptide-His6 constructs) and similar core peptide structure we anticipate that formation of the correct ring topologies is very likely.

To further confirm that wild-type cycle topologies are preserved in the context of MBP fusions, we established new constructs and produced bioactive nisin (precursor peptide NisA modified by NisBC enzymes) and lactacin 481 (chimeric precursor peptide ProcA3.3-LctA modified by ProcM) as C-terminal fusions to MBP. The antimicrobial activity of these peptides relies on enzymatic modification and formation of the correct cycle topologies, which is demonstrated in the new Supplementary Fig. 3. The Supplementary method section has been updated accordingly and the full sequences were added to Supplementary Table 5.

The following sentence has been added to the main text (p.6, lines 19-23): “To further demonstrate that even more complex wild-type thioether cycle topologies are correctly introduced in the context of precursor peptides fused to the C-terminus of MBP, we expressed the lantibiotics nisin and lactacin 481 as MBP fusions and confirmed their antimicrobial activity (Supplementary Fig. 3).”

3. Regarding the phage display/biopanning experiments of ProcM-modified libraries (Figure S3 and Figure 4). Authors show that ProcM co-expression is needed for peptides to bind uPA, clearly demonstrating that the peptides need to be post-translationally modified in order to bind target. However, no data on the cyclization state of the peptides is presented. It is implied that the peptides are cyclized by the comparison with previously generated S-S linked, phage displayed peptide libraries that bind to uPA, but the cyclization status of the peptides in the current study is not explicitly shown. The authors should report ETD data on the best-hit binders shown in Figure 4 to see if they are indeed cyclized. Cyclization will be critical for in vivo stability of any peptide biologic developed by this approach. Thus, it is highly important to demonstrate that ProcM is generating cyclized peptides in this assay.

Reply:

We agree with the Reviewer and have performed additional analysis using chemical and enzymatical modifications and MS/MS demonstrating the extent of lanthionines in each of the uPA- and streptavidin-binding peptides (new Supplementary Figure 5 and Supplementary Tables 9-15). The former Supplementary Figure S3, which only showed dehydration, was removed.

4. Why is +/- ProcM data for PEP335 not shown in 4c?

Reply:

We initially limited Figure 4c to the peptide with higher affinity to uPA identified in the same library (library 1) and also skipped to show the cognate analytics. The binding curve for PEP335 has now been added to Figure 4c (new colors were used to provide better overview) and also analytical data is presented.

5. Table S8. What is being reported? Average mass? This should be stated somewhere. Also, the errors seem to be a little high (many are ~ 30 ppm or more). This instrument should be capable of single digit or less ppm errors. Could the authors comment on this?

Reply:

Table S8 (now Supplementary Table 6, due to additional tables added to the Supplementary section) showed the average mass for all peptides >10 kDa. For peptide PEP226 (< 10 kDa) the monoisotopic mass is shown. This has been clarified in a subscript of new Supplementary Table 6.

Our routine mass spectra for identification of the lanthipeptides were deconvoluted with a limited number of iterations for sake of efficiency. This appeared to be admissible since the sequence of every peptide was known and consequently a mass deviation below 1 Da was considered to be insignificant.

We now reprocessed all spectra of peptides > 10 kDa with an increased number of iterations. For all peptides the mass deviation of the measured mass from the calculated mass was found to be ≤ 0.1 Da. (approx. 10 ppm). The mass spectra in Fig. 3b and the data in new Supplementary Table 6 were replaced for the data from the reprocessed spectra.

Please note that ms data of all phage selected peptides were removed from Supplementary Table 6. These peptides were re-analyzed with high accuracy during the revision process and data are shown in Supplementary Tables 9-15.

Reviewer #2

a) The authors should demonstrate the presence and, ideally, also the functional importance of the thioether linkage(s) in the sequence hits isolated against uPA and streptavidin (Fig. 4). The MS data of Fig S3 show the occurrence of Ser/Thr dehydration but no experimental evidence is provided to support the presence and number of thioether linkage(s) in these compounds. Several data actually argue against it. First, the data of Fig. 2 show that ProcM-catalyzed thioether formation in the presence of dehydrobutyrine is inefficient. Fig. 4e-f data show a decrease in streptavidin binding affinity for peptide PEP330 and PEP331 in the presence of reductant, suggesting that these peptides are cyclized via a disulfide bridge, not via a thioether linkage. The same experiment (+/- reductant) was not performed with the uPA binding peptides (Fig.4c), contributing to the uncertainty regarding the presence (and importance) of thioether linkages in these compounds.

Reply:

Please see reply to comment b) as well, which addresses parts of this comment.

The reduced cyclization efficiency in the presence of dehydrobutyrine in Fig. 2 is indeed significant. However, this artificial monocyclic peptide is not the most efficient substrate and cyclization efficiency will depend on the sequence, and preformed structure of the precursor peptide and the Lan enzymes used. That efficient cyclization from dehydrobutyrine frequently occurs can be seen in many natural lanthipeptides and some of our selected peptides (e.g. PEP331, Supplementary Table 15), which we characterized now in depth.

The decreased binding of the streptavidin specific peptide under reducing conditions is modest in our opinion (EC50s are quite similar) and was performed to demonstrate that putative disulfides do not contribute to the binding affinity. The modest decrease in binding could rather be explained by

unwanted effects such as reduction of disulfides in the detection antibody. Since the uPA protein itself contains 12 disulfides we anticipated major structural rearrangements under reducing conditions and did not perform the binding experiment under such conditions at the time, since a loss of peptide binding was anticipated due to uPA unfolding. This assumption was confirmed during the revision of the manuscript, since we observed loss of catalytic activity of the uPA protein under reducing conditions.

b) The structure of at least one of the bioactive peptides isolated against each target should be unambiguously elucidated. Hit elucidation is far from being trivial in the present system. Take the strongest consensus sequence isolated against uPA (PEP332). Fig S3a shows that the major product carries three dehydrated S/T out of 4 possible ones. Considering all possibilities with respect to the target AA for dehydration, plus all the possible thioether connectivities of a putative monocyclic and bicyclic product, there are >30 possible structures. This issue is further complicated by the fact that thioether formation with dehydroalanine (from dehydration of Ser) can occur even in the absence of enzymatic catalysis (as shown by data in Fig. 2a), thus generating two possible stereoisomers for each product containing such a bridge.

Reply:

We performed additional peptide analytics to address this important comment that was also similarly raised above (Reviewer #1, comment 3). Our new results show that both streptavidin-binding peptides contain two efficiently formed non-overlapping lanthionines. The same efficiently formed lanthionine configuration was observed in PEP332, whereas PEP333, PEP334, and PEP335 contain mixtures of one or two lanthionines. The data is provided in the new Supplementary Figure 5 and Supplementary Tables 9-15.

c) On the related note, it remains unclear if multiple products are actually formed after ProcM processing of the sequences described in Fig. 4a and 4d. The MS data of Fig S2 provide information only on the number of dehydrated amino acids, which may or may not be cyclized via a thioether bond. Thus, it is possible that multiple regio- and/or diastereomeric products could be formed in these reactions. HPLC or LC-MS traces of these reactions should be provided in the SI to clarify this point. This aspect should be commented upon in the text.

Reply:

Since we performed an in depth analysis of the selected peptides during the revision process, which shows and discusses the different peptide species, we believe the recorded HPLC traces can be omitted. The traces are however attached to this response letter as Appendix Figure 2.

Other points:

a) The results of Fig. 2a-b show that the impact of S->T substitution on preventing ProcM-catalyzed cyclization is substantial. The same comparison should be made for the sequences fused to the C-terminus of the pIII coat protein (panels c and d of Fig. 2). This information is particularly important considering that all hit sequences contain one or more Thr within the target sequence.

Reply:

We agree with the reviewer that the impact of S->T substitution on preventing ProcM-catalyzed cyclization in this particular model peptide is substantial and anticipated the same reduced

cyclization efficiency in the context of a C-terminal pIII on phage. Indeed, we performed this experiment in the ProcM enzymatic system (not the NisBC enzymatic system) and received the predicted results (see Appendix Fig. 3). However, the cyclization efficiency will depend on the sequence, the enzymatic system used, and the preformed structure of the precursor peptide. Our objective here was to use a model peptide that helps us to identify productive ways to achieve display on phage, which it did. Our manuscript shows clearly that efficient cyclization can occur from threonine (and the dehydrated dehydrobutyrine) and we provide excellent tools for the future development of scaffolds for improved library designs with increased cyclization efficiency. We therefore believe that the reduced cyclization efficiency of this particular threonine-containing model peptide is not relevant for the overall finding described in the manuscript and can be omitted.

b) The impact of lanthipeptide display on the infectivity of the phage particles should be measured and commented upon. While the authors provide information regarding the original size of the libraries, the chemical diversity that remains effectively accessible by the method is also dictated by how many phage particles can be propagated after panning. This information can be gained from simple control experiments.

Reply:

To address this comment the following sentence has been added to the method section (p.17, lines 22-25): "The number of transducing units (t.u.) present in the established phage libraries was determined by infectivity titration on susceptible *E. coli* TG1F+ cells. Phage titers were in the range of 10^{14} t.u. ml⁻¹ (> 10.000-fold library size) and no sign of reduced infectivity caused by C-terminally displayed peptides was noted."

It is further mentioned in the method section that phages were diluted to $\sim 1.5 \times 10^{13}$ t.u. for the first round of selection which equals >1.000 fold original library size.

c) The SI carries a section about peptide synthesis (page 21) but no synthetic peptides were discussed in the main text.

Reply:

Synthetic peptides were used to establish the Factor Xa reporter assay to monitor peptide cyclization in Figure 1 and are discussed in the main text and the Figure legend 1.

Reviewer #3

However, the authors stop short of demonstrating that the peptides that they have derived exhibit activity. This could easily be tested with the peptides that bind to uPA, since this is a protease that can be assayed for activity in the presence or absence of the binding peptides to determine whether the peptides inhibit activity. It would be very worthwhile to determine the potency of the inhibitory activity of at least the highest affinity lanthipeptide for uPA, and to compare this to previous disulfide constrained peptides reported in the literature.

Reply:

As suggested by Reviewer #3, we determined the inhibitory activity of selected peptides on uPA in a fluorogenic substrate assay. The results are shown in the new Supplementary Figure 4. The Supplementary method section has been updated accordingly.

The following sentence has been added to the main text section (p.7, lines 14-15): “Furthermore the selected peptides inhibited the catalytic activity of uPA according to their binding profile and in a ProcM-dependent manner (Supplementary Fig. 4).”

On p. 8, line 22 “...and inhibition.” has been added.

Additional changes:

Analytical results of peptides selected by phage display, which have been generated during the revision process, are now mentioned and briefly discussed in the main text in new sections p.7, lines 15-22; p.8, lines 5-8 and lines 22-28; and p.9, lines 1-5.

Due to additional Supplementary Figures and Tables, the previous numbering of Supplementary Figures and Tables has changed at some points and new references to Supplementary Figures and Tables were added. These changes are highlighted.

In the Acknowledgment section (p. 25, lines 16-17) we added some more people who were instrumental during the revision of the manuscript.

In Supplementary Fig. 2c we noticed some copy-paste errors in the stated proton counts. These were corrected.

In the author contribution section R.R. and G.N.M. have been added to analyzing data (p.25, line 23).

In the Competing financial interests section the initials of two contributors (p. 25, line 28) were reversed and have been corrected.

In Figure legend of Figure 4d and Figure 4e the peptide identifiers were mixed up. Figure 4d shows the binding of PEP330 (not PEP331) and Figure 4e shows the binding of PEP331 (not PEP330). This mistake has been corrected. Accordingly, in the main text of the manuscript the peptide identifiers have been corrected (p. 7, line 27-28 was changed to “While PEP331 was fully dehydrated, one position in PEP330 had largely escaped dehydration...” and p8, line 1 to “PEP330 bound streptavidin regardless of ProcM co-expression...” and on p. 8, line 2-3 was changed to “In contrast, the binding of PEP331 to streptavidin was strictly dependent on post-translational modifications...”).

In the Figure 3 legend (p. 27, line 5) “for full sequences see Supplementary Table 5” has been added.

A data availability statement has been added to the Methods section (p 21, lines 9-12).

A Statistical analysis subsection has been added to the Method section (p.17, lines 15-19).

On p.2, line 5, p. 3, line 15, and p.4, line 7: “thioether bridged” was replaced by “thioether-bridged”

On p.2, line 26: a comma was introduced “Unfortunately,...”

On p.18, line 19: “Analysis of peptides by mass spectrometry” has been changed to “Analysis of MBP-fused prochlorosin peptides by mass spectrometry”. Please note that all additional peptide ms analytics performed during the revision process was carried out on different equipment (by co-author Rick Rink), since co-author Tobias Aumüller left the company in the meantime.

Appendix Figure 1

a

b

c

Appendix Figure 1 ELISA-based screening for thioether formation in model peptides containing a FactorXa cleavage site and different residues flanking the serine and cysteine residues in the core sequence. (a) Schematic drawing of the precursor peptides used in the screen. All peptides had an OmpA signal sequence and a linker containing a free cysteine (red) upstream of the NisA leader. The core peptides contained a FactorXa cleavage site (FXa) flanked by affinity tags (FLAG, His₆) as indicated. The serine and cysteine, relevant for thioether formation, in the core peptide are colored and x (black boxes) indicate flexible positions as shown in the graph below. (b) Precursor peptides (core sequences are indicated) were expressed with or without NisBC, captured from cell lysates, and subjected to FXa-digestion and ELISA detection. The protease resistance relative to untreated (no FXa) samples was calculated and data representing mean \pm s.d. of three independent cultures analyzed in duplicate is shown (c) As in (b), but only peptides showing the highest degree of FXa resistance (red box) as indication for thioether formation were analyzed again and an extra washing step with 25 mM DTT after peptide capture onto the ELISA plate was included to reduce putative disulfides that might otherwise lead to false positive FXa cleavage resistance. The core sequence ASWIEGRVC showed very high FXa cleavage resistance (indication for thioether formation) and was selected for further experiments.

Appendix Figure 2

Appendix Figure 3 UPLC profiles of phage-selected lanthipeptides. The data were collected by using a gradient from 5 to 95% acetonitrile in water with 0.1% formic acid over 10 min at a flow rate of 0.3 ml/min on an Acquity UPLC Protein BEH C18 column (2.1 mm x 50 mm, Waters). The time course of elution was detected with Total Ion Current (TIC) Intensity.

Appendix Figure 3

Appendix Figure 2 Assessment of the cyclization status of artificial lanthipeptides displayed on the C-terminus of phage pIII. As in Fig. 2d, but including a S77T mutant, which results in reduced, yet significant cyclization.

REVIEWERS' COMMENTS:

Reviewer #1 (Remarks to the Author):

The revised manuscript satisfactorily addresses each of my concerns. This study will make an important contribution to the field of natural product engineering, and the authors should be commended for tackling such a difficult, yet important problem. The study is appropriate for Nat. Commun. and will be very impactful.

Reviewer #2 (Remarks to the Author):

The revised version of the manuscript addresses most of the points raised by the reviewers. In response to concerns regarding the unambiguous structural identification of the selected lanthipeptides, the authors have included data describing the major product of cyclization. At the same time, the HPLC traces provided as Appending Figure 3 in the Rebuttal Letter clearly show that multiple products are produced in these reactions as suggested by Reviewer #2. This is a very important piece of information that should be disclosed in the paper as it shows a potential limitation of the method. In addition to the emphasizing the value of the technology as done throughout the paper, it is important to outline also potential drawbacks associated with it. Accordingly, this reviewer requests that Appending Figure 3 is included as part of the SI and a short comment referring to the formation of multiple products in the cyclization reactions is included in the text. The best place for that seems to be the first paragraph of the Discussion section where details of the post-translational maturation process are discussed. After this revision, the manuscript will be suitable for publication in this journal. Congratulations to the authors for the excellent work.

Reviewer #3 (Remarks to the Author):

The authors have addressed my concerns and the paper is suitable for publication.